# Human Activity Recognition by Sequences of Skeleton Features

**DOI:** 10.3390/s22113991

**Published:** 2022-05-25

**Authors:** Heilym Ramirez, Sergio A. Velastin, Paulo Aguayo, Ernesto Fabregas, Gonzalo Farias

**Affiliations:** 1Escuela de Ingeniería Eléctrica, Pontificia Universidad Católica de Valparaíso, Av. Brasil 2147, Valparaíso 2362804, Chile; heilym.ramirez@pucv.cl (H.R.); paulo.morales.a@mail.pucv.cl (P.A.); 2School of Electronic Engineering and Computer Science, Queen Mary University of London, London E1 4NS, UK; sergio.velastin@ieee.org; 3Department of Computer Science and Engineering, Universidad Carlos III de Madrid, 28903 Madrid, Spain; 4Departamento de Informática y Automática, Universidad Nacional de Educación a Distancia, Juan del Rosal 16, 28040 Madrid, Spain; efabregas@dia.uned.es

**Keywords:** fall detection, activity recognition, machine learning, human skeleton, images sequence

## Abstract

In recent years, much effort has been devoted to the development of applications capable of detecting different types of human activity. In this field, fall detection is particularly relevant, especially for the elderly. On the one hand, some applications use wearable sensors that are integrated into cell phones, necklaces or smart bracelets to detect sudden movements of the person wearing the device. The main drawback of these types of systems is that these devices must be placed on a person’s body. This is a major drawback because they can be uncomfortable, in addition to the fact that these systems cannot be implemented in open spaces and with unfamiliar people. In contrast, other approaches perform activity recognition from video camera images, which have many advantages over the previous ones since the user is not required to wear the sensors. As a result, these applications can be implemented in open spaces and with unknown people. This paper presents a vision-based algorithm for activity recognition. The main contribution of this work is to use human skeleton pose estimation as a feature extraction method for activity detection in video camera images. The use of this method allows the detection of multiple people’s activities in the same scene. The algorithm is also capable of classifying multi-frame activities, precisely for those that need more than one frame to be detected. The method is evaluated with the public UP-FALL dataset and compared to similar algorithms using the same dataset.

## 1. Introduction

The detection and recognition of human activities have attracted the attention of researchers around the world in recent years. In this research field, fall detection using machine learning techniques has become very important due to the aging of the population in developed countries and the increase in the cost of hospitalizations [1]. For that reason, this kind of application has been growing due to the implementation of safety measures [2] in high-risk work environments, shopping malls, hospitals, nursing homes [3], etc.

These solutions have been implemented both for closed private places and in open public spaces, so there is an essential difference between indoor and outdoor scenarios.

On the one hand, in controlled indoor spaces, such applications use sensors, such as gyroscopes, accelerometers, inertial sensors, barometers, etc., for fall detection. Such sensors can be embedded in smartphones, wearable necklaces, or smart bracelets. They can also be placed directly on the waist or chest of the person [4]. However, they have the disadvantage that the devices must be attached to the subjects’ bodies. This can be uncomfortable and is not always feasible because these types of sensors must be worn constantly and in many cases present battery and wireless connection problems [5,6,7].

On the other hand, outdoor approaches may perform human activity detection and recognition through video images. The use of video images has some advantages over the wearable sensor-based approaches: (a) users do not need to wear any sensors, (b) the approach can be implemented in open spaces and not only in a laboratory environment, and (c) there can be more than one person in the scene [8]. Many works dedicated to human activity detection from video cameras can be found in the literature. In some cases, they use pose detection through a human skeleton as a feature extraction method. These approaches allow the detection of various types of activities such as walking, running, jumping, jogging, or falling, among others. They also enable the detection of more than one person in the same scene, which is a substantial advantage over other approaches [9]. For example, in [10] the authors present a network-based fall detection in videos, where experiments show a high sensitivity of 98.46% on the multiple cameras fall dataset and 100% on the UR-Fall [11] dataset.

In this context, we previously presented [12] a vision-based approach for fall detection and activity recognition using human skeleton estimation for feature extraction. It can detect several subjects in the same scene, so it uses a filtering method to detect the activity of a single person of interest, discriminating the skeleton of the other subjects. It can also be used in real open and uncontrolled environments, where a large number of people can be present at the same time in the scene, for example, in a shopping mall. The method has been evaluated with the public databases UP-Fall [13] and UR-Fall [11]. In both cases, the results outperformed other systems using the same dataset [14], achieving a 98.84% accuracy and an *F*_1_-Score of 97.41%, which performs fall detection using a k-nearest neighbor (KNN) classifier. This system uses pose detection of the human skeleton as an input feature.

Despite the very good results obtained, some activities were not well recognized because more than one frame is required to identify them accurately. This implies that the activity cannot be classified instantaneously, so it needs more time to be detected. That is, the activity starts at an initial state and requires intermediate steps before it reaches the final state so that it can be properly recognized as an activity. Therefore, a method that uses only one frame to detect this type of activity is not adequate. Several articles related to this topic have been published recently. For example, in [15], the authors present a deep neural network approach consisting of an automatic encoder followed by a CNN, where the model is trained with image sequences. In [16], the authors propose a method for human fall detection with CNNs. The system uses the concept of “dynamic imaging”, which combines the frames of a video into a single image to identify action sequences. In [17], the authors propose a multi-camera system for video-based fall detection. They augment the estimation of human pose (OpenPifPaf algorithm) by supporting multi-camera, multi-person tracking and an LSTM neural network to predict classes. The method achieves an *F*_1_ score of 92.5% on the UP-Fall dataset.

To be able to detect activities that cannot be detected with a single frame, some modifications are made to our previous works. The result is what is presented here, in which the main contribution is the development of a method capable of detecting those activities that cannot be recognized using a single frame. The major effort is made in the feature extraction stage rather than the machine learning methods, which shows a better performance in comparison with previous works. Regardless of the classification method, in all cases the results are improved. In this approach, the feature extraction method seeks to obtain the pose of the subject to which the activity is identified and for which a skeleton detection algorithm that has delivered excellent results in previous work is used [12]. This method has been proven with different datasets and machine learning algorithms showing better results than previous works [18,19,20]. Therefore, the main contributions of this work are:The use of several frames of a video to recognize an activity is proposed. This approach allows us to correctly detect those activities that require a time greater than the period required to capture a frame. The use of several frames of a video is proposed to recognize an activity. Note that this approach allows performing two kinds of classification problems: bi-classification (fall/not fall), and multi-classification (recognition of more than two activities). The proposed method provides better results than those reported in previous works, including those activities that were recognized with only one frame. That is why the method represents a global improvement in the detection of activities.This approach differs from many existing works in that the effort is made in the feature extraction stage by proposing to use skeleton features to estimate the human pose in a frame. Unlike previous works, the feature vector is formed by combining skeleton features from several consecutive frames. In this paper, we describe a study to specifically determine the frames needed to detect activities.To show the improved detection performance, the approach is validated using different ML methods to build an activity classifier. Better results are obtained for most of the machine learning methods used.Finally, the robustness and versatility of the approach have been validated with two different datasets, achieving in both cases better results compared to those previously reported in the literature.

The rest of the paper is organized as follows. Section 2 presents some fall detection approaches that can be found in the literature. In addition, the UP-Fall database, which will be used later to test the developed algorithm, is briefly described. Section 3 and Section 4 describe the developed algorithm. Section 5 shows the machine learning models and evaluation metrics used. Section 6 shows the experimental results and a comparison with previous results for the UP-Fall dataset. Finally, Section 7 summarizes the main conclusions and future work.

## 2. Fall Detection Datasets and Related Work

A considerable number of works related to activity detection with sensors can be found in the literature. Many of them present some drawbacks such as frequent battery replacement, high costs, use of wearable devices, sensing distance, computational load, etc. Several recent works have focused on human posture detection from video images to avoid some of these drawbacks. These approaches can be divided into two groups: (a) works that process video images to detect activities using artificial intelligence techniques (SVN, CNN, YOLO, etc.) directly from those images; (b) research that uses human skeleton feature detection (e.g., obtained with OpenPose, AlphaPose, etc.) in video images to classify activities, also with artificial intelligence techniques to improve the results. The use of skeleton detection can improve the training time effectively as well as eliminate the effects of blurriness, light, and shadows. Some of those activities cannot be classified using only one frame. That is why some recent research has focused on activity detection from image sequences.

Table 1 shows a summary of the state of the art of works related to activity detection. The first column is the reference to each article. The second column indicates whether the approach is able to detect multiple activities or not and how many. The third column shows whether the paper uses an algorithm for pose detection with skeleton detection. The fourth column indicates whether the developed algorithm uses image sequences to classify the activities. The fifth column indicates whether the method uses depth sensors for detection. The sixth column shows the artificial intelligence algorithm used in each case. The last column shows the dataset used to perform the tests.

As can be seen, 63% of the reviewed literature uses skeletons for activity detection, and the most used method is OpenPose. Only 20% of the works can detect multiple activities. The most used public datasets for testing are: UR-Fall (30%), UP-Fall (16%), and own datasets (16%), while 70% of the articles report the use of sequences for activity detection.

The most commonly used database is UR-Fall, and one of the articles using this database is [21] in which the authors present enhanced dynamic optical flow. Their experimental results showed that the fall detection accuracy improved about 3% and the processing time between 40 and 50 ms.

In the application presented in [22], the authors propose a deep learning fall detection framework for a mobile robot. This method runs three times faster than YOLOv3-tiny on a Raspberry Pi without any hardware accelerator and with good detection performance.

In [23], the authors presented a dense block-based drop detection method with a multi-channel convolutional fusion (MCCF) strategy. This method shows excellent results for fall detection over the same dataset (F-score of 0.973).

In [24], the authors developed an architecture to classify human fall events using 2D CNN and GRU. Experimental results with this dataset show that the proposed model obtains an accuracy of 99%. Although the proposal has a good performance, it has the disadvantage of using the depth information of the databases with which they test and validate their approach. This makes fall detection easier to detect and more difficult to apply since it is more common to use RGB images in real environments.

In [25], the authors proposed a method capable of detecting human falls in video sequences using multi-channel CNN and SVM. The results are competitive with those obtained by the state of the art on the UR-Fall dataset.

In [34], the authors proposed methods that apply OpenPose for real-time multi-person 2D pose estimation and movement. These methods were tested with RNN and LSTM models to learn the changes in human joint points in continuous time. Experiments over UR-Fall show that they can detect falls with a recognition accuracy of up to 98.1% and 98.2%, respectively.

The work presented in [37] describes a vision-based fall tracking method where upper body joints are grouped into one segment to increase the fall classification ratio. This method can be beneficial to achieve efficient tracking of human activities and provide a useful technique to distinguish falls from other daily life activities.

In Table 1, it can be seen that the works [43,44,45] are the most similar to the present work. The characteristics to be taken into account are: (a) fall detection, (b) recognition of different activities, and (c) the use of skeletons and a sequential frame analysis methodology, resulting in good performance (accuracy over 90%). It is important to note that the present work cannot be properly compared with the three mentioned works due to the fact that the training and validation datasets are different (The present work uses the UP-Fall and UR-Fall datasets to compare directly with [12]). On the other hand, the datasets used in [44,45] (FFD and TST-Fall, respectively) have images with depth information from the use of kinect cameras, which makes the method much easier to apply. However, it is not very practical to use non-conventional depth cameras. Based on the above, the present work uses a novel method in which datasets acquired from conventional cameras are considered. In addition, a greater number of activities are recognized than almost all the works mentioned in the review.

On the other hand, Table 2 shows the most used datasets for fall detection (reflected in Table 1). The first column names and references each dataset. The second column shows the fall type included in each dataset. The third column shows other kind of activities (different from falling) included in the dataset. The fourth column indicates the number of attempts or sequences performed for each activity and fall. Finally, Columns 5 and 6 show the methods used by the original authors to evaluate their dataset and the performance obtained with each method. It should be noted that Table 2 shows the performance obtained by the original authors of each dataset, while Table 1 shows the performance obtained in the most recent works that use the same datasets.

## 3. Methodology of the Proposed Approach

The present work focuses on improving the performance of fall detection and activity recognition using video data. The main hypothesis is that the results obtained in [12] can be improved by carrying out a sequential analysis of frames, instead of analyzing and classifying each frame independently, even using the same machine learning models and the same datasets.

A sequence of several frames is used for the recognition of an activity instead of detecting the activity with a single frame. This allows the correct detection of activities that require more time than the period required for the capture of a single frame, thus improving the performance in those activities that previously were not recognized with a single frame. Therefore, the proposed method represents an overall improvement in the detection of activities.

Here, the use of skeletal features to estimate the human pose in a frame focuses the effort on the feature extraction stage, differentiating from the works in the state of the art. Unlike previous works, the feature vector is formed by combining the skeletal features of several consecutive frames. We describe a study to specifically determine the number of frames needed to detect an activity. The approach is validated using various ML methods for activity recognition, obtaining better results for most of the machine learning methods used. Finally, the robustness and versatility of the approach is validated with two different datasets, obtaining in both cases superior results compared to those previously reported in the literature.

The proposed methodology for fall detection and activity recognition is shown in Figure 1. The proposed method is divided into two parts: feature extraction and activity recognition using different machine learning models. The process begins with the collection of video images, followed by the extraction of features by estimating the pose of the human skeleton, and then the selection of sliding time windows for training machine learning models that predict the fall or activity for each time window.

### 3.1. Selection of Sliding Windows

The novelty of the proposal includes the combination of consecutive skeleton features by considering image frame sequences through a sliding windows approach (SW). The design of these windows is shown in Figure 2. The process consists of defining the size of the windows in seconds, so that each window will cover the number of frames equal to the fps of the video times the size of the window (SWL=fps×seconds). So, for an 18 fps video with 1 s windows, each window will span 18 frames (SWL=18). Finally, each window scrolls one frame to the right of the previous one until one window spans the last frame of the video.

The total number of sliding windows can be determined by Equation (Equation 1), as follows:(1)$$SW_{m}=\#FramesVideo-SWL+1,$$
where #FramesVideo is the total number of frames of the video.

Each window spans an SWL number of frames, and each frame contains the activity of one person. Following the methodology proposed in [12], each frame will have the 51 skeleton characteristics associated with the person’s pose and the label associated with the activity performed in that frame. This means that each sliding window will contain SWL skeletons with their respective associated tag. The size of the sliding window can be determined by (1,SWL,51).

### 3.2. Feature Vector Construction

Once the sliding windows are designed, the goal is to predict the activity corresponding to each window. To do that, one skeleton is obtained for each frame of the sliding window. After that, each skeleton is labeled using a machine learning model. Finally, the statistical mode is used to output the most frequent label of the sliding window. Thus, each sliding window has a single tag/label associated with the activity performed in that time interval. To build the features vector to train and validate the ML models and taking into account that each window has a size in three dimensions, it is necessary to resize each window size to two dimensions, so each window will have the following shape: (1,SWL×51).

The first 51 columns correspond to the features of the first skeleton, and the last 51 columns correspond to the 51 features of the last skeleton. Finally, the feature vector and the label vector will have the form:(2)$$(SW_{m}\times \#Videos,SWL\times 51),$$
and the size of labels vector is:(3)$$(SW_{m}\times \#Videos,1),$$
where #Videos varies depending on the number of videos in the dataset.

The following section shows the parameters selected in the construction of this vector and an optimization proposal to correctly detect all activities.

## 4. Study of Feature Vector Parameters Settings

As in [12], the estimation of the human pose is performed using AlphaPose [52], available at https://www.mvig.org/research/alphapose.html (accessed on 20 January 2022). AlphaPose is an open-source method to detect the pose of one or more subjects in the scene. Its methodology consists of taking RGB images as input and performing pose detection with a model previously trained with the COCO dataset. For each subject detected in the image, there is a set of 17 key points or joints with coordinates (x,y) that compose a skeleton. In addition, it provides an individual joint score *s* for each key point and also, an overall detection score for the (17 × 3) attributes. The features are obtained to train a classifier that detects falls and recognizes activities. In this way, a sequence of RGB images or a video is converted into a sequence of joints and scores that form the features. Vector shells are used to learn to distinguish the different actions.

### 4.1. Exhaustive Search

An objective function is defined as: recall maximization as a function of window size (*W*), number of skeletons (*S*), and number of features (*F*):(4)Max_Recall(W,S,F)≈100%

Taking into account how a skeleton is obtained from the pose of a person, there are some restrictions to consider:Videos or images must have only one person; so if there is more than one person in the scene, only the characteristics of the person of interest are used, and the skeletons of the other people are discriminated.The duration of the video must be longer than the window size (*W*).The duration of the activity must be longer than or equal to the window size (*W*).When obtaining the skeleton features for each video frame, the key-points must always be 17 per person.

Once the images without people have been eliminated and the filter designed in [12] is applied (which selects only the skeleton of the person of interest in the images that contain more than one person), the size of the window (*W*) must be greater than 0 s and less than or equal to T seconds: 0<W≤T. Therefore, for this work, three window sizes were selected: 0.5, 1, and 2 s. Taking into account that for each frame of the window there is a skeleton, the feature vector can be optimized using all the skeletons of the window or less; then, the number of skeletons (*S*) in the window can be less or equal to the frames per second of the video (fps) times the size of the window.
(5)1<S≤(fps×W)
(6)1<S≤SWL

### 4.2. Data Acquisition

To test the hypothesis that the use of temporary sliding windows can improve results in [12], the UP-Fall database [13] was selected for a more direct comparison.

Data acquisition was performed using the same methodology described in [12], skeleton features of the person pose were extracted for each frame of the videos in the UP-Fall dataset. The videos in UP-Fall have 18 fps (frames per second).

The experiments were performed using the UP-Fall dataset, which was divided using 70% for training and 30% of the data for testing. Four classification models were validated: random forest (RF), support vector machine (SVM), multilayer perceptron (MLP), and K-nearest neighbor (KNN). For each model, 10 rounds of cross-validation were carried out.

The exhaustive search technique was selected to find the candidate that optimizes the feature vector by reducing the number of skeletons per window without sacrificing system performance. This exhaustive search was implemented with Matlab 2020b (Windows OS) and Python 3.6 (training and validation).

### 4.3. Skeletons Selection

Given that UP-Fall videos are 18 fps, that the maximum number of skeletons *S* per window *W* is fps×W and that the number of features is 51×S, then:For W=0.5 s: 2≤S≤fps×0.5.For W=1 s: 2≤S≤fps×1.For W=2 s: 2≤S≤fps×2.

An example of the selection of skeletons per sliding window is shown in Figure 3. Considering a window of 1 s duration, the exhaustive search of two skeletons up to fps×W or SWL skeletons (*S*) per window was performed.

The selection of the skeletons for each window is defined by the following code:
for ws in {0.5, 1.0, 2.0} do    for s in {1...FPS} do        feature_vector = new Matrix{ frames_of_video.size \        - (ws ∗ FPS) + 1 , 51 ∗ (s+1) }        frames = new Vector{ s+1 }        for x in {0...s} do            frames[x] = (x / s) ∗ ((ws ∗ FPS) - 1)        frames[s] = (ws ∗ FPS) - 1        i = 0        while (i <= (frames_of_video.size - (ws ∗ FPS))) do            f = 0            for frame in frames do                feature_vector[i][f] = frames_of_video[frame+i]                f++            i++
where:
$S_{0}$: Initial position of the skeleton.$S_{i}$: Position of the skeleton to be selected.*F*: The data number of the feature vector (51×S).

Finally, a feature vector was obtained for each possible candidate with which the ML models are trained and validated. Each vector has the form:(7)$$\left(\left(SW_{m}\times \#Videos\right),F\right)$$

Size of labels vector:(8)$$\left(\left(SW_{m}\times \#Videos\right),1\right)$$

### 4.4. Best Solution

The exhaustive search of the $Max_{R}ecall$ was carried out taking into account 10 cross-validations for each possible W and S parameter.

From the results obtained by the exhaustive search, the recall obtained for each candidate pair W and S is plotted in Figure 4. This graph shows the maximization of the recognition recall of the 12 UP-Fall activities when using sliding windows with skeleton sequences instead of performing the frame-by-frame analysis proposed in [12].

Finally, the candidate with the highest recall for the entire system is selected:Recall = 96.43%.Window size (W) = 2 s.Number of skeletons for window (S) = 3.Number of features (F) = 153.

As can be seen in Figure 4, the next best candidates are found in the 2 s window with 18 and 35 skeletons. It is possible to conclude that the highest recall values are obtained with 2 s windows. Since the exhaustive search is performed with the UP-Fall database, using three skeletons works well for this database, it may be different from other databases. On the other hand using three skeletons and choosing the first frame, the middle frame, and the last frame makes sense to recognize an action that occurs in the full size of the window. We believe that this is caused by the relative simplicity of the model, which is not prone to overfitting and curse of dimensionality.

## 5. Metrics and Associated Parameters

As in [12], the skeleton sequences are preprocessed to eliminate empty frames and because some images show other people in addition to the volunteer performing the action to be recognized, the skeleton with the highest overall score is chosen. Thus, the other skeletons are eliminated from the training process.

### 5.1. Models

The results of the proposed approach were validated by using the same experimental methodology described in [12]. The experiments were performed using the UP-Fall dataset, selecting 70% of the data for training and 30% of the data for testing. total of 10 rounds of cross-validation were performed with five classification methods: random forest (RF), support vector machine (SVM), multilayer perceptron (MLP), K-nearest neighbor (KNN), and AdaBoost.

### 5.2. Metrics

This work uses the same performance metrics used in [12] for a direct comparison:(9)$$\mathrm{accuracy}=\frac{\mathrm{TP}+\mathrm{TN}}{\mathrm{TP}+\mathrm{FP}+\mathrm{TN}+\mathrm{FN}}\;$$
(10)$$\mathrm{precision}=\frac{\mathrm{TP}}{\mathrm{TP}+\mathrm{FP}}\;$$
(11)$$\mathrm{recall}=\frac{\mathrm{TP}}{\mathrm{TP}+\mathrm{FN}}\;$$
(12)$$\mathrm{specificity}=\frac{\mathrm{TN}}{\mathrm{TN}+\mathrm{FP}}\;$$
(13)$$F_{1}=2\times \frac{\mathrm{precision}\times \mathrm{recall}}{\mathrm{precision}+\mathrm{recall}}\;,$$
where:TP (True positives) = “fall” detected as “fall”;FP (False positives) = “not fall” detected as “fall”;TN (True negatives) = “not fall” detected as “not fall”;FN (False negatives) = “fall” detected as “not fall”.

## 6. Experimental Results

The results obtained from the proposed methodology are shown below.To verify that the recall and the performance of the system, in general, can be maximized when using sliding windows with skeleton sequences, all the experiments in the present work were carried out using the methodology proposed in Section 3 and the parameters selected from Section 4.

### 6.1. Fall Detection and Activity Recognition Using an LSTM

Firstly, to test deep learning models, an activity recognition experiment was performed using the same feature vector described in Section 3 and Section 4, for which an LSTM (long short-term memory) neural network was designed, taking into account that the proposed approach uses temporary window sliders with the skeleton sequences.

The structure used to determine the classification of feature sequences obtained in the human pose detection phase has its nucleus in an LSTM network. The proposed structure is composed of 51 input features for the LSTM with 100 hidden states, followed by a fully-connected layer with 100 nodes, and finished by a softmax layer with 12 nodes. These final nodes are related to the previous job classes [12], namely 12 activities in total: falling forward on hands, falling forward on knees, falling backwards, landing sitting on an empty chair, falling from the side, walking, standing, pick up an object, sit, jump, lie down, and kneel. The first five activities correspond to human fall actions, and the next seven correspond to simple daily human activities. The selected loss function was cross-entropy loss with the Adam optimizer with a learning rate of 0.01.

It is necessary to clarify that the input to the network is a sequence of features extracted from a temporal sequence of frames in which the feature set is related to the highest pose estimation score detected in just one frame, i.e., the algorithm just detects one person per frame.

The LSTM network was trained and validated using the UP-Fall database. A total of 10 rounds of cross-validation with 70:30 partitions was performed. Finally, the results obtained for the recognition of the 12 UP-Fall activities are: accuracy = 81.14%, precision = 27.76%, recall = 31.82% and $F_{1}$-score = 29.53%. The low performance of the network was attributed to the imbalance present in the 12 classes of the UP-Fall dataset. It is possible that with more balanced datasets the performance of the network increases, even using the same methodology proposed in this work (sequences of skeletons).

Based on the poor results obtained with the LSTM neural network, it was decided to test the proposed method with well-known machine learning models; so the experiment was divided into two parts: fall detection (bi-classifier system) and activity recognition (multi-classifier system).

### 6.2. Fall Detection with UP-Fall

Fall detection was performed by means of binary classifiers. The input data was the feature vector described in Section 3 and Section 4. There were 220,660 frames, with which 203,525 sliding windows were created of which 43,469 (21.36%) windows corresponded to sequences with falls and 160,056 (78.64%) to sequences without falls.

As in [12], the original 12 class labels were converted so that all five types of dropouts were encoded as “dropout” and the rest as “no dropout”. The five classification models were trained separately (random forest, support vector machine, multi-layer perceptron, K-nearest neighbor and AdaBoost) and validated by 10 rounds of cross-validation (k-fold = 10) using 70:30 random partitions of the entire dataset.

The results obtained by each classifier are shown in Table 3 (performance of the proposed method). The best results were obtained with RF that achieved an accuracy of 99.81%, a recall of 99.81% and an $F_{1}$-score of 99.56%. Table 3 compares the results obtained from the proposed method with those obtained in [12]. It is possible to observe that the use of sliding windows leads to improved performance for four out of the five classification models.

The best confusion matrix for each of the five classification models is shown in Figure 5. The classifier with the best performance is RF in which, of the 4,425 fall data, only 5 (0.11%) were not recognized as such.

Figure 6 compares the best model (RF: Windowing) of the proposed method with the best model (RF: Frame by Frame) of the method proposed in Ramirez et al. [12]. It can be seen from the two confusion matrices that even with the same datast and the same type of classification model, the model trained with sliding windows shows better performance, increasing the percentage of fall detection from 98.89% to 99.89%. As mentioned in the previous sections, the difference between the two methods is that the present work performs the detection of falls by means of sequences of frames, whereas in [12], detection is carried out independently frame by frame.

Table 4 compares our performance to other studies performing fall detection with camera view skeleton sequences for fall detection and using the same database: UP-Fall. It is possible to observe that the performance of our proposal exceeds the performance delivered by the work carried out in [12,17,32]. Moreover, unfortunately, the works in [17,32] do not implement activity recognition.

Therefore, it is possible to conclude that our hypothesis is proven, by showing that it is possible to improve the results obtained with the same classification models proposed in [12] that uses a single modality (RGB images) and the same dataset, along with human skeletal sequences.

### 6.3. Activity Recognition with UP-Fall

As described in Martínez-Villaseñor et al. [13], the UP-Fall dataset contains 12 different types of actions of which the first 5 correspond to fall activities, falling forward with hands, falling forward with knees, falling backward, landing sitting on an empty chair, and fall sideways, and the next 7 correspond to human daily activities, walking, standing, picking up an object, sitting, jumping, lying down, and kneeling down.

Activity recognition is carried out using multi-class classifiers. As in the fall detection case, the input data is the feature vector described in Section 3 and Section 4, but in this case, each sliding window contains 1 of the 12 UP-Fall activities. Of the 203,525 sliding windows, 1120 (0.55%) correspond to the activity falling forward with hands, 1102 (0.54%) to falling forward with knees, 1384 (0.68%) to falling backward, 1109 (0.54%) to landing sitting on an empty chair, 1371 (0.67%) to picking up an object, 38,787 (19.06%) to falling sideways, 50,997 (25.06%) to walking, 45,871 (22.54%) to standing, 1331 (0.65%) to sitting, 21,786 (10.7%) to jumping, 37,383 (18.37%) to lying down, and 1284 (0.63%) to kneeling down. Finally, following the same methodology used in [12], the dataset is randomly split 10 times into 70% for training and 30% for testing. The five models are validated separately (RF, SVM, MLP, KNN and AdaBoost) and validated through 10 rounds (k-fold = 10) of cross-validation.

Table 5 compares the results obtained in this work with the results obtained in Ramirez et al. [12]. It is observed that the performance of all the models of the proposed method exceeds the performance of all the models in Ramirez et al. except for the SVM. It is also possible to observe that our best model outperforms the best model in Ramirez et al.

Figure 7 compare thes confusion matrix of our best model (RF) versus the best model in Ramirez et al. (Figure 8) [12] (RF). From the confusion matrix of our best model, it is possible to observe that the performance of the classifier increases and that the system is more accurate in recognizing each activity, managing to increase accuracy from 99.78% to 99.92% and recall from 88.97% to 96.03%. Our best model is capable of recognizing falling activities with more than 90% accuracy and activities such as sitting, jumping, and kneeling with 100%.

One of the biggest drawbacks in [12] is that when performing an analysis of independent frames, the falling activities (first five activities) are often confused with other activities such as lying down (Activity 11). This can be seen in the matrix of confusion of Figure 8. Therefore, our original hypothesis that it is possible to improve the performance of classification models for activity recognition and avoid confusion between classes by using skeleton sequences by means of optimized sliding windows is demonstrated (Section 4).

As Table 6 shows, our method outperforms results from similar work using camera-view skeleton sequences for fall detection and activity recognition such as Wang et al. [43], Zhu et al. [44] and Yin et al. [45] where they reach an accuracy of 95%, 99.04%, and 93.9%, respectively, while we achieve an accuracy of 99.91%. It is important to point out, as mentioned in the Section 2, the present work cannot be adequately compared with the three mentioned works because the training and validation datasets are different. The present work uses the UP-Fall and UR-Fall datasets to compare directly with [11]. Furthermore, the works of Zhu et al. [44] and Yin et al. [45] use images with depth information, which makes the method much easier to design but not very practical because of the need to have depth cameras. The latter proves the value of our method since, even though it is more complex to develop, better results are achieved even using a single conventional camera. It is also important to highlight the notorious imbalance of the 12 classes of the UP-Fall database. Even so, our proposal manages to achieve high performance with each ML model when it comes to recognizing any of the 12 activities.

### 6.4. Fall Detection with UR-Fall

To check the efficiency and robustness of the proposed method, the experiments were repeated with another dataset, UR-Fall [11]. UR-Fall has 9800 frames with which 5712 sliding windows are created, of which 1870 (32.74%) correspond to falls and 3842 (67.26%) to not-falls. As with the UP-Fall database, the same five fall detection classification models are trained and validated with 10 rounds of cross-validation using 70:30 partitions.

Table 7 compares the results obtained with the proposed method versus the results obtained in Ramirez et al. [12]. Even though the performance of the [12] fall detection system using UR-Fall was already quite good, it can be seen that by using sliding windows with skeleton sequences it is possible to increase the performance of the fall detection system. As Table 7 shows, our best model (RF) outperforms the best model in Ramirez et al.

The confusion matrix of our best model (RF) using UR-Fall is shown in Figure 9 (left). Here, it is possible to observe that the model can detect falls with 100% precision. Therefore, based on the results obtained, the hypothesis is demonstrated that by using optimized sliding windows (Section 4) with skeleton sequences, it is possible to increase the performance of fall detection systems even when using other datasets.

Table 8 compares our performance to other studies performing fall detection with camera view using a skeleton for fall detection and using the same database, UR-Fall. It is possible to observe that the performance of our proposal exceeds the performance delivered in all the works cited in the table.

Finally, Table 9 shows the execution time for 10 cross-validations of each ML model. All models were trained and validated on a Notebook with an Apple M1 Chip (8 cores) and 16GB RAM. On average, each model took 0.59 s considering the creation of the Jason file with the AlphaPose skeleton data, and 0.07 s for reading the file and making the prediction for each frame.

From Table 9, it is possible to conclude that the KNN model is the best option considering its high accuracy rates and its fast execution performance.

## 7. Conclusions

This paper has proposed and tested a method for fall detection and activity recognition from RGB video sequences estimating the pose of a person in the entire video sequence. The feature vector is implemented by means of skeleton features that contain information about the pose of the person contained in the video. The proposal was evaluated using the UP-Fall database through five machine learning models (RF, SVM, MLP, KNN and AdaBoost), with which a good performance of the method was demonstrated that surpassed other fall detection systems and activity recognition referenced in the state of the art.

The main contribution of this work is that by using skeletons, it is possible to represent the joints of the human body, correctly estimating the pose of a person, which allows detecting a fall or correctly recognizing a daily activity. In addition, the use of time sequences of these skeletons makes it possible to significantly reduce the confusion between classes by correctly recognizing very similar activities. On the other hand, it is shown that by designing the feature vector from skeleton sequences, it is possible to reduce the amount of data by using only some of the skeletons without decreasing system performance.

For future work, we intend to test the proposal on real videos, since it has only been tested on datasets created in the laboratory and under controlled environments. Also, the construction of an activity recognition system of several people in the same video needs to be considered. In addition, the development of an algorithm for tracking one or more people in different scenes will be considered. Such an algorithm could recognize all the activities carried out and search or create datasets for activity recognition with multiple people and multiple cameras in different scenes.

## Figures and Tables

**Figure 1 sensors-22-03991-f001:**
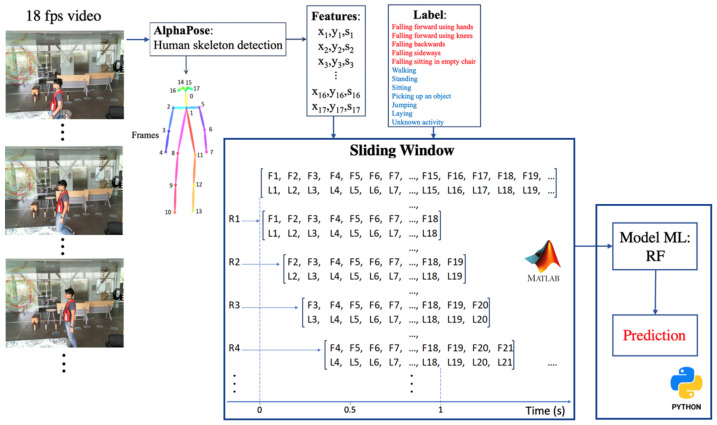
Workflow for activity recognition.

**Figure 2 sensors-22-03991-f002:**
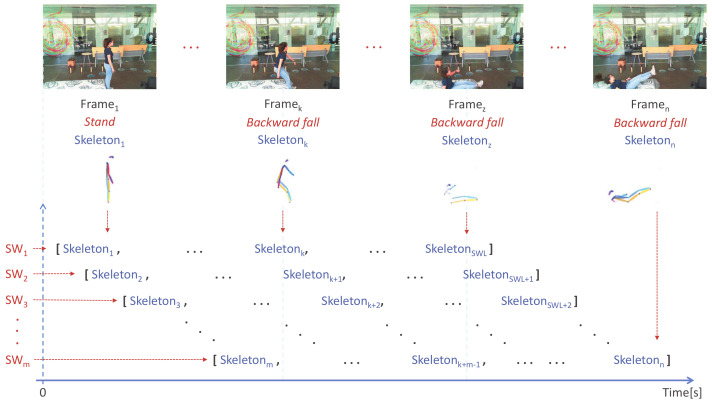
Sliding Windows for activity recognition.

**Figure 3 sensors-22-03991-f003:**
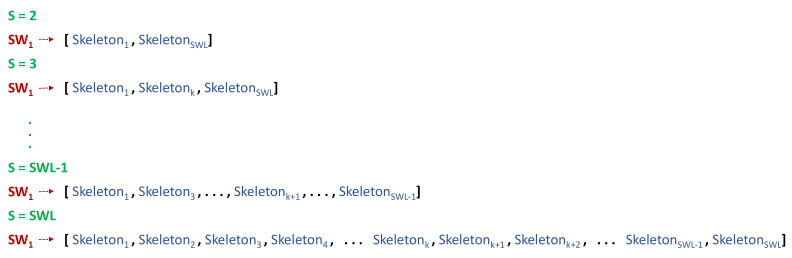
Selection of number of skeletons per window.

**Figure 4 sensors-22-03991-f004:**
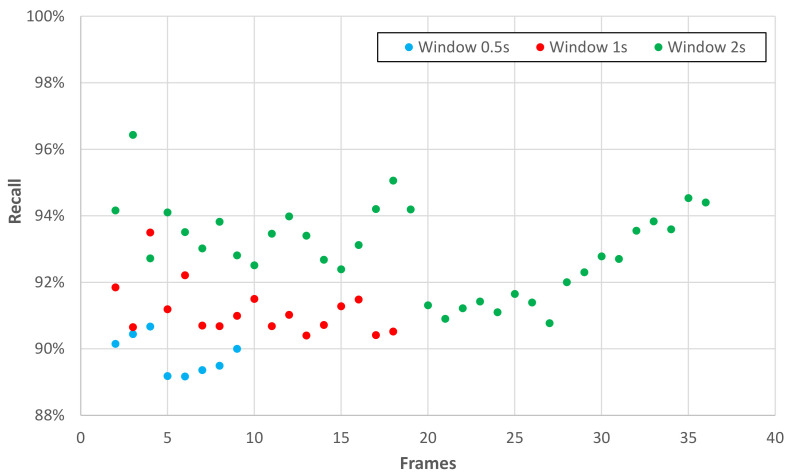
Exhaustive search graph.

**Figure 5 sensors-22-03991-f005:**
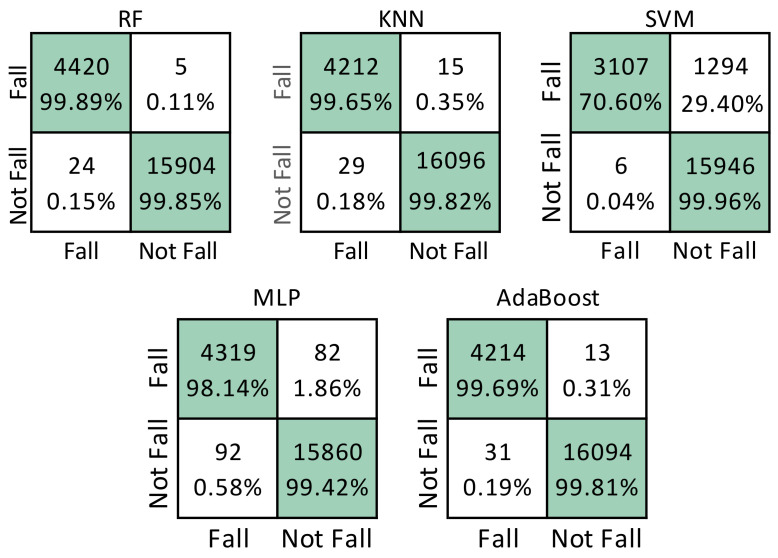
Confusion matrix for fall detection for RF, KNN, SVM, MLP, and AdaBoost, respectively. True positives (TP) are in the upper left corner; false positives (FP) are in the lower left corner; false negatives (FN) are in the upper right corner; true negatives (TN) are in the lower right corner.

**Figure 6 sensors-22-03991-f006:**
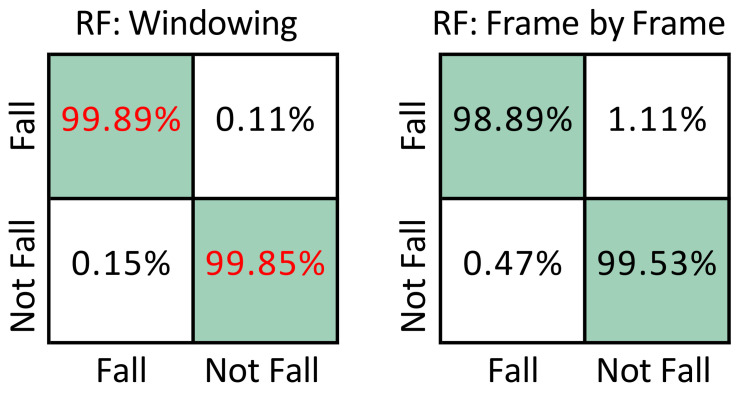
Comparison of confusion matrices for fall detection using UP-Fall: RF proposed method (left) vs. Frame by Frame+RF (right) [12]. Best results are shown in red.

**Figure 7 sensors-22-03991-f007:**
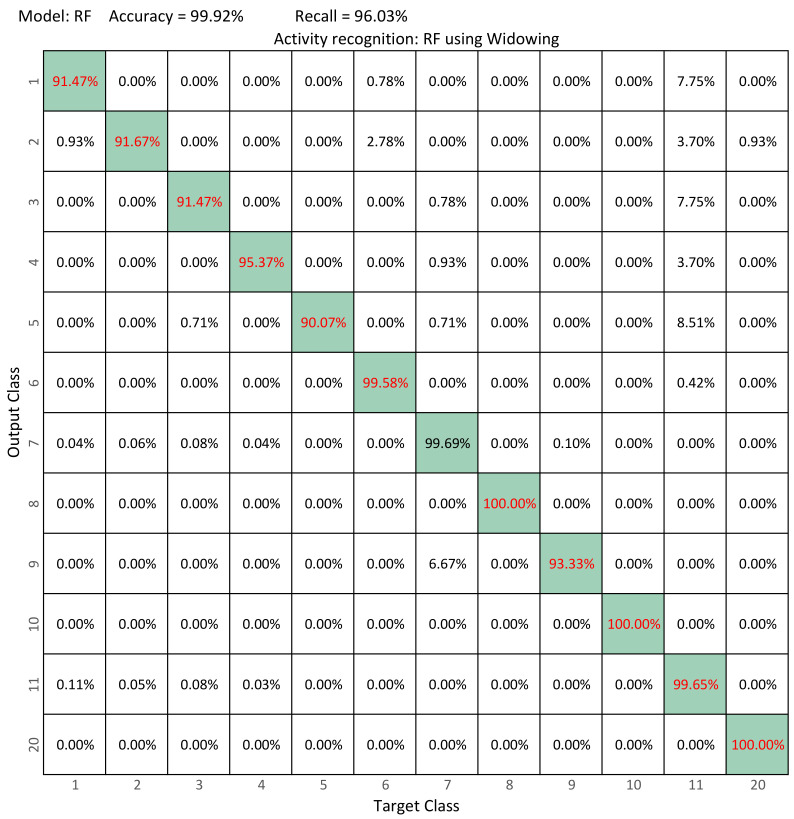
Confusion matrices in activity recognition: Windowing+RF. Best results are shown in red.

**Figure 8 sensors-22-03991-f008:**
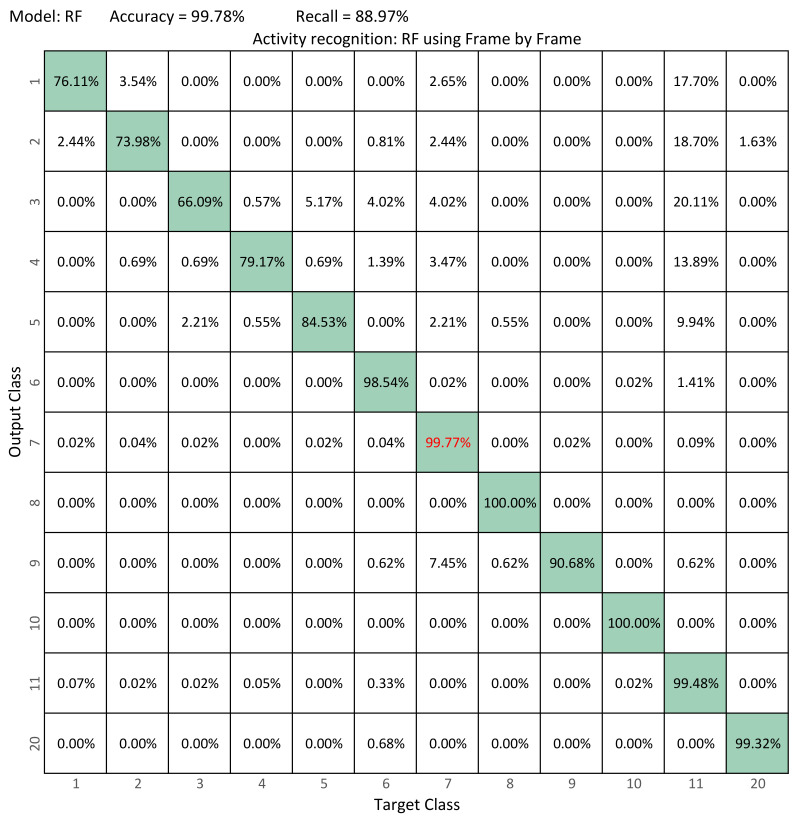
Confusion matrices in activity recognition: Frame-by-Frame+RF. Best results are shown in red.

**Figure 9 sensors-22-03991-f009:**
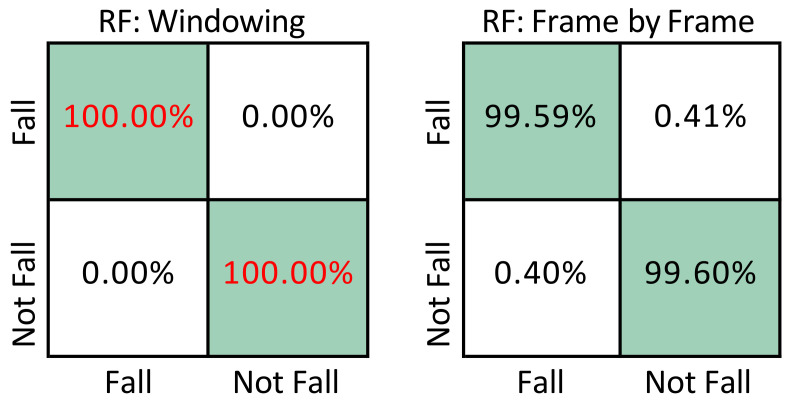
Comparison of confusion matrices for fall detection using UR-Fall: RF proposed method (**left**) vs. Frame-by-Frame+RF (right) [12]. Best results are shown in red.

**Table 1 sensors-22-03991-t001:** State of the art for human activity recognition by video images. Please consider the following meaning ✓—Yes, X—No.

Ref	Multi Activity	Skeleton	Sequence	Cam RGB	Model	Dataset
[21]	X	X	✓	✓	CNN	UR-Fall
[22]	X	X	✓	✓	NanoDet-Lite	UR-Fall
[23]	X	X	✓	✓	MCCF	UR-Fall
[24]	X	X	✓	Depth	2D CNN-GRU	UR-Fall
[25]	X	X	✓	Depth	CNN-SVM	UR-Fall
[26]	X	X	✓	Depth	RVM	own
[27]	X	X	✓	Depth	ST-GCN	TST-Fall
[28]	X	X	X	✓	CNN-YOLO	CMD-Fall
[29]	X	X	X	✓	CNN	own
[30]	X	X	X	✓	KNN-SVM	BOMNI
[31]	X	X	X	✓	CNN	own
[17]	X	own	✓	✓	LSTM	UP-Fall
[32]	X	own	✓	✓	AutoEncoder	UP-Fall
[10]	X	OpenPose	✓	✓	LSTM	UR-Fall
[33]	X	PoseNet	✓	✓	GRU	UR-Fall
[34]	X	OpenPose	✓	✓	LSTM-GRU	UR-Fall
[21]	X	OpenPose	✓	✓	SVM	UR-Fall
[35]	X	OpenPose	✓	✓	SVM	UR-Fall
[36]	X	OpenPose	✓	✓	LSTM	CMU
[37]	X	PoseNet	✓	✓	CNN	own
[38]	X	PoseNet	✓	✓	CNN-RNN	own
[39]	X	OpenPose	✓	Depth	RF	SDU-Fall
[40]	X	own	✓	✓	GRU	SDU-Fall
[41]	-	OpenPose	✓	-	-	-
[14]	X	AlphaPose	X	✓	KNN	UP-Fall
[42]	✓(5)	X	X	✓	DAG-SVM	own
[43]	✓(7)	Yolo v3	✓	✓	3D CNN	PKU-MMD
[44]	✓(4)	OpenPose	✓	Depth	DNN	FDD
[45]	✓(8)	own	✓	Depth	MC-LSTM	TST-Fall
[12]	✓(12)	AlphaPose	X	✓	RF-SVM MLP-KNN	UP-Fall

**Table 2 sensors-22-03991-t002:** Vision-based datasets for fall detection.

Dataset	Fall Types	Other Activities	Trials	ML Method	Performance
SDUFall [46]	Fall to the floor	Sitting, walking, squatting, lying, bending	6 actions 10 times	Bag of words model built upon curvature scale space features	Accuracy: 79.91%, Sensitivity 81.91%, Specificity 76.62%
SFU-IMU [47]	15 types of falls	Walking, Standing, Rising, Ascending stairs, Picking up an object	3 repetitions	SVM	Sensitivity 96%, Specificity 96%
UR-Fall [11]	From standing, from sitting on a chair	Lying, walking, sitting down, crouching down	70 sequences	SVM	Accuracy: 94.99%, Precision 89.57%, Sensitivity 100%, Specificity 91.25%
CMD-FALL [48]	While walking, lying on the bed, sitting on the chair	Horizontal movement	20 actions	CNN: Res-TCN	*F*_1_-Score (Activity): 39.38%, *F*_1_-Score (Fall): 76.06%
Fall-Dataset [49]	Fall to the floor	Standing, sitting, lying, bending and crawling		CNN	Accuracy: 74%
PKU-MMD [50]		Drinking, waving hand, putting on the glassed, hugging, shaking...	6 sequences	RNN SVM LSTM	$F_{1}$-Score: 52.6% 13.1% 33.3%
K-Fall [51]	15 types of falls	21 types of activities		Conv-LSTM	Accuracy: 99.32%, Recall: 99.3%
UP-Fall [13]	Forward using hands, forward using knees, backward, sideward, sitting	Walking, standing, sitting, picking up an object, jumping, laying, kneeling down	3 repetitions	RF SVM MLP KNN	Accuracy: 32.33% 34.40% 27.08% 34.03%

**Table 3 sensors-22-03991-t003:** Performance (mean ± standard deviation) obtained for each model of the proposed fall detection system using UP-Fall. The best results are shown in bold.

**Performance Ramirez et al. [12]**					
**Model**	**Accuracy (%)**	**Precision (%)**	**Recall (%)**	**Specificity (%)**	**$F_{1}$-Score (%)**
RF	**99.34** ± **0.03**	**98.23** ± **0.17**	**98.82** ± **0.10**	**99.48** ± **0.05**	**98.52** ± **0.08**
SVM	98.81 ± 0.07	98.15 ± 0.19	96.50 ± 0.27	99.47 ± 0.05	97.32 ± 0.17
MLP	97.39 ± 0.10	93.87 ± 0.85	94.57 ± 1.15	98.21 ± 0.29	94.21 ± 0.27
KNN	98.84 ± 0.06	97.53 ± 0.15	97.30 ± 0.24	99.29 ± 0.04	97.41 ± 0.16
**Performance of the Proposed Method**					
**Model**	**Accuracy (%)**	**Precision (%)**	**Recall (%)**	**Specificity (%)**	**$F_{1}$-Score (%)**
RF	**99.81** ± **0.04**	**99.30** ± **0.17**	**99.81** ± **0.07**	**99.81** ± **0.05**	**99.56** ± **0.09**
SVM	93.37 ± 0.15	99.76 ± 0.05	69.12 ± 0.80	99.95 ± 0.01	81.66 ± 0.57
MLP	98.95 ± 0.14	97.62 ± 0.49	97.47 ± 0.86	99.35 ± 0.14	97.54 ± 0.33
KNN	99.69 ± 0.04	99.17 ± 0.18	99.39 ± 0.12	99.77 ± 0.05	99.28 ± 0.10
AdaBoost	99.71 ± 0.04	99.11 ± 0.14	99.52 ± 0.11	99.76 ± 0.04	99.31 ± 0.10

**Table 4 sensors-22-03991-t004:** Comparison with other methods using skeletons for fall detection with UP-Fall. Please note that ✓—yes, X—No.

Methods	Dataset	CAM	Skeleton Sequences	Accuracy
Taufeeque et al. [17]	UP-Fall	RGB	✓	98.28%
Galvão et al. [32]	UP-Fall	RGB	✓	98.62%
Ramirez et al. [12]	UP-Fall	RGB	X	99.34%
**Our method**	**UP-Fall**	**RGB**	✓	**99.81%**

**Table 5 sensors-22-03991-t005:** Performance (mean ± standard deviation) obtained for each model of the activity recognition system using UP-Fall.

**Performance in Ramirez et al. [12]**					
**Model**	**Accuracy (%)**	**Precision (%)**	**Recall (%)**	**Specificity (%)**	**$F_{1}$-Score (%)**
RF	99.45 ± 1.02	**96.60** ± **0.48**	**88.99** ± **0.56**	99.70 ± 0.50	**92.34** ± **0.39**
SVM	**99.65** ± **0.01**	93.85 ± 0.65	87.29 ± 0.83	**99.79** ± **0.01**	90.20 ± 0.59
MLP	98.93 ± 0.17	85.39 ± 1.69	71.44 ± 2.30	99.34 ± 0.11	75.95 ± 1.84
KNN	99.60 ± 0.01	91.65 ± 0.55	84.17 ± 0.81	99.76 ± 0.01	87.35 ± 0.63
**Performance of the Proposed Method**					
**Model**	**Accuracy (%)**	**Precision (%)**	**Recall (%)**	**Specificity (%)**	**$F_{1}$-Score (%)**
RF	**99.91** ± **0.01**	**97.73** ± **0.28**	**95.60** ± **0.39**	**99.95** ± **0.01**	**96.63** ± **0.33**
SVM	98.60 ± 0.04	95.60 ± 0.67	57.40 ± 0.60	99.14 ± 0.02	62.87 ± 0.81
MLP	99.28 ± 0.17	82.71 ± 2.23	78.97 ± 2.01	99.58 ± 0.10	79.89 ± 1.96
KNN	99.81 ± 0.01	92.49 ± 0.40	91.50 ± 0.37	99.89 ± 0.01	91.95 ± 0.35
AdaBoost	99.81 ± 0.03	95.53 ± 0.50	92.56 ± 0.38	99.89 ± 0.02	93.97 ± 0.39

**Table 6 sensors-22-03991-t006:** Other methods using skeletons for fall detection and activity recognition. Please consider the following meaning ✓—Yes, X—No.

Methods	Dataset	CAM	Activities	Skeleton Sequences	Accuracy
Wang et al. [43]	PKU-MMD	RGB	7	✓	95.00%
Zhu et al. [44]	FDD	Depth	4	✓	99.04%
Yin et al. [45]	TST-Fall	Depth	8	✓	93.90%
Ramirez et al. [12]	UP-Fall	RGB	12	X	99.65%
**Our method**	**UP-Fall**	**RGB**	**12**	✓	**99.91%**

**Table 7 sensors-22-03991-t007:** Performance (mean ± standard deviation) obtained for each model of the proposed fall detection system using UR-Fall. Best results are shown in bold.

**Performance in Ramirez et al. [12]**					
**Model**	**Accuracy (%)**	**Precision (%)**	**Recall (%)**	**Specificity (%)**	**$F_{1}$-Score (%)**
RF	**99.11 ± 0.43**	**99.18 ± 0.59**	97.53 ± 1.81	**99.71 ± 0.21**	**98.34 ± 0.80**
SVM	98.60 ± 0.30	96.50 ± 0.88	98.37 ± 0.88	98.69 ± 0.32	97.42 ± 0.60
MLP	90.79 ± 4.14	86.63 ± 13.60	83.06 ± 11.61	93.69 ± 8.15	83.19 ± 5.55
KNN	98.88 ± 0.31	98.41 ± 0.96	97.41 ± 1.11	99.43 ± 0.33	97.90 ± 0.60
AdaBoost	98.95 ± 0.31	98.42 ± 0.98	**97.67 ± 1.12**	99.43 ± 0.34	98.04 ± 0.59
**Performance of the Proposed Method**					
**Model**	**Accuracy (%)**	**Precision (%)**	**Recall (%)**	**Specificity (%)**	**$F_{1}$-Score (%)**
RF	**99.51** ± **0.33**	**99.35** ± **0.68**	99.15 ± 0.71	**99.69** ± **0.32**	**99.25** ± **0.51**
SVM	96.39 ± 0.92	90.60 ± 2.17	**99.36** ± **0.55**	94.94 ± 1.34	94.77 ± 1.23
MLP	92.18 ± 4.71	88.53 ± 8.62	89.42 ± 17.04	93.39 ± 5.82	87.39 ± 10.58
KNN	99.28 ± 0.39	98.88 ± 0.64	98.95 ± 0.84	99.45 ± 0.31	98.91 ± 0.58
AdaBoost	99.42 ± 0.34	99.25 ± 0.63	98.99 ± 0.80	99.64 ± 0.30	99.12 ± 0.52

**Table 8 sensors-22-03991-t008:** Comparison with other methods using skeletons for fall detection with UR-Fall. Please consider the following meaning ✓—Yes, X—No.

Methods	Dataset	CAM	Skeleton Sequences	Accuracy
Guan et al. [10]	UR-Fall	RGB	✓	99.00%
Kang et al. [33]	UR-Fall	RGB	✓	99.46%
Lin et al. [34]	UR-Fall	RGB	✓	98.20%
Chhetri et al. [21]	UR-Fall	RGB	✓	95.11%
Dentamaro et al. [35]	UR-Fall	RGB	✓	99.00%
Ramirez et al. [12]	UR-Fall	RGB	X	99.11%
**Our method**	**UR-Fall**	**RGB**	✓	**99.51%**

**Table 9 sensors-22-03991-t009:** Execution times for 10 k-fold of cross validations for each ML model.

Training and Validation Times [s]			
Model	Fall Detection with UP-Fall	Activity Recognition with UP-Fall	Fall Detection with UR-Fall
RF	1644.32	2262.06	23.54
SVM	64,696.15	120,464.13	31.04
MLP	981.78	4913.33	317.17
KNN	376.60	439.88	1.23
AdaBoost	3148.76	3936.27	49.81

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
