# Peer review of "Human Activity Recognition by Sequences of Skeleton Features"

_sensors, 2022, doi:10.3390/s22113991_

Round 1
Reviewer 1 Report
The paper deals with human activity recognition from ordinary videos, with the main emphasis on the detection of falls. It extends previous research where state-of-the-art NN techniques were applied to extract skeleton poses from individual video-frames, and the skeletons were then fed to various ML techniques for action recognition. In the current paper, sequences of skeletons are considered instead of individual poses, which increases the fall recognition accuracy over two experimental datasets (UP-Fall and UR-Fall).
Strengths:
- The topic of human activity recognition from ordinary videos is highly relevant.
- The state-of-the-art review is comprehensive (I especially appreciate Table 1)
- The basic idea of using skeleton sequences instead of individual poses for action recognition is sane.
Weaknesses:
- The proposed method based on sliding windows is quite trivial: instead of a single pose, the authors add additional poses to the ML input. The only slightly advanced idea is to select a limited number of poses from each sliding window. However, I did not understand how the parameters W and S were selected. What is the recall that is maximized by exhaustive search? It is quite surprising that the optimal result was obtained for just 2 skeletons per SW. Some analysis of this behaviour would be good. Was this setting used in all the experiments in Section 6?
- No real comparison to state-of-the-art is made except for a comparison with authors’ own previous work that worked with isolated poses. It is no surprise that the sequences work better – there is more information available. The authors compare their results to several other works in Section 6.3, but a comparison of accuracy over different datasets makes no sense. On the other hand, relevant works that use sequences of poses over the same dataset (e.g, [9], [16]) are not mentioned in the comparison.
- I find it extremely suspicious that a LSTM model evaluated by the authors worked poorly. Other works have found LSTM models very good on similar tasks. Without a strong explanation of this behaviour, I am more inclined to think that the reported LSTM implementation has some problems. Actually, I believe that LSTMs (or potentially other recursive NN models) are the strongest competitors of the proposed approach and should be thoroughly discussed in the paper.
- The costs of ML training with the sequences of skeletons are never considered.
Additional comments:
- The text needs to be thoroughly revised. Apart from a paragraph in Spanish (middle of page 2), there are many repetitions of the same information (e.g., the list of contributions from page 3 is very much repeated on page 7).
- It is vital to clearly explain the task that is being studied – is it action detection (finding a specific action, e.g. fall, within a long motion sequence/stream) or action recognition (deciding type of a short motion piece that was manually segmented)? How is the input motion processed? Sliding windows are used, so multiple windows will be evaluated per one input. How are the results combined?
- Give more attention to explaining the basic principles of the proposed methods, instead of delving fast into detailed descriptions.
- Table 2 is not referenced from text. What does it show? First few columns deal with dataset properties, the last two columns mention ML methods and performance. What methods are these? Clearly not the best existing approaches for given datasets (e.g., the accuracy stated for UP-Fall is very low).
- The code for selection of skeletons within the SW is confusing. The idea, as far as I understand it, is to take a given number of evenly spaced poses from each SW. In the code, you have confusing notation (S_i vs. S-1 vs. S_i(1,i))
Author Response
Response to Reviewer 1 Comments
Comments: The paper deals with human activity recognition from ordinary videos, with the main emphasis on the detection of falls. It extends previous research where state-of-the-art NN techniques were applied to extract skeleton poses from individual video frames, and the skeletons were then fed to various ML techniques for action recognition. In the current paper, sequences of skeletons are considered instead of individual poses, which increases the fall recognition accuracy over two experimental datasets (UP-Fall and UR-Fall).
Response: Thank you for these comments.
Strengths:
- The topic of human activity recognition from ordinary videos is highly relevant.
- The state-of-the-art review is comprehensive (I especially appreciate Table 1)
- The basic idea of using skeleton sequences instead of individual poses for action recognition is sane.
- Author response: Thank you very much for these comments.
Response: Thank you for these comments.
Weaknesses:
Point 1: The proposed method based on sliding windows is quite trivial: instead of a single pose, the authors add additional poses to the ML input. The only slightly advanced idea is to select a limited number of poses from each sliding window. However, I did not understand how the parameters W and S were selected.
Response 1: Thank you for this comment. The window size (W) and the number of skeletons (S) are selected from the best candidate found in the exhaustive search, which is located at W=2 and S=3. This is explained in subsection 4.4.
Point 2: What is the recall that is maximized by an exhaustive search?
Response 2: Thank you for this comment. The recognition recall of the 12 UP-Fall activities is maximized by using sliding windows with skeleton sequences instead of performing the frame-by-frame analysis proposed in [12]. This is explained in subsection 4.4.
Point 3: It is quite surprising that the optimal result was obtained for just 2 skeletons per SW. Some analysis of this behaviour would be good.
Response 3: Please note that the best result is obtained with 3 skeletons. Please also note that since the exhaustive search is performed with the UP-Fall database, using 3 skeletons works well for this particular case, and it may be different for other databases.
Point 4: Was this setting used in all the experiments in Section 6?
Response 4: Yes, we have used the same setting in all experiments. We have added a comment in section 6.
Point 5: No real comparison to state-of-the-art is made except for a comparison with authors’ previous work that worked with isolated poses. It is no surprise that the sequences work better – there is more information available. The authors compare their results to several other works in Section 6.3, but a comparison of accuracy over different datasets makes no sense.
Response 5: Thank you very much for the remark. The idea is to show other works that have managed to recognize actions with skeleton sequences. We have modified the text to clarify this point in Table 6. On the other hand, using 3 skeletons and choosing the first frame, the middle frame and the last frame makes sense to recognize an action that occurs in the window. The fact that the system works better with less data is also related to the concept of the Curse of dimensionality (see [*, **] below). This is clarified in subsection 4.4.
* Verleysen, M., & François, D. (2005, June). The curse of dimensionality in data mining and time series prediction. In International work-conference on artificial neural networks (pp. 758-770). Springer, Berlin, Heidelberg
** Bengio, Y., Delalleau, O., & Le Roux, N. (2005). The curse of dimensionality for local kernel machines. Techn. Rep, 1258, 12.ISO 690
Point 6: On the other hand, relevant works that use sequences of poses over the same dataset (e.g, [9], [16]) are not mentioned in the comparison.
Response 6: Thank you for this remark. We have modified the text to add this analysis in section 6.2 and Table 4. It is possible to see that our approach outperforms other works. The comparison has been done with [16] (now [17)) and [32]. The reference [9] (now [10]) uses UR-FALL instead of UP-FALL such as our work and [17,32,12].
Point 7: I find it extremely suspicious that an LSTM model evaluated by the authors worked poorly. Other works have found LSTM models very good on similar tasks. Without a strong explanation of this behaviour, I am more inclined to think that the reported LSTM implementation has some problems. I believe that LSTMs (or potentially other recursive NN models) are the strongest competitors of the proposed approach and should be thoroughly discussed in the paper.
Response 7: We do agree with the reviewer, it was also a surprise for us that the LSTM didn't work well. Several tests were carried out for the recognition of multiple activities. Different LSTM configurations with various densities and hyperparameters were tried. We hypothesise that there is too much-unbalanced information, which normally causes overfitting in the models. For example see classes 1, 2, 3, 4, 5, 9 and 12, which contain less than 1% of the total 203.525 SWs as mentioned in the second paragraph of section 6.3.
Point 8: The costs of ML training with the sequences of skeletons are never considered.
Response 8: Thank you very much for this remark. The costs of training ML models with the skeleton sequences were added at the end of section 6 (see Table 8).
Weaknesses:
Point 9: The text needs to be thoroughly revised. Apart from a paragraph in Spanish (middle of page 2), there are many repetitions of the same information (e.g., the list of contributions from page 3 is very much repeated on page 7).
Response 9: Thank you very much for this suggestion. We have revised the whole article.
Point 10: It is vital to clearly explain the task that is being studied – is it action detection (finding a specific action, e.g. fall, within a long motion sequence/stream) or action recognition (deciding the type of a short motion piece that was manually segmented)? How is the input motion processed? Sliding windows are used, so multiple windows will be evaluated per one input. How are the results combined?
Response 10: The focus of the work is activity recognition as indicated in the title. The purpose is to recognize different actions in a video that lasts up to 4 minutes. All frames of the video are labeled. One sliding window (SW), which contains 3 frames, is associated with the most frequent (statistical mode) of the three labels. This is explained in section 3.2.
Point 11: Give more attention to explaining the basic principles of the proposed methods, instead of delving fast into detailed descriptions.
Response 11: Thank you for this remark. We think that the principles of the approach are explained in detail in Figure 1 (section 3). However, we have added a new sentence to clarify this point in section 3.
Point 12: Table 2 is not referenced from the text. What does it show? First, a few columns deal with dataset properties, the last two columns mention ML methods and performance. What methods are these? not the best existing approaches for given datasets (e.g., the accuracy stated for UP-Fall is very low).
Response 12: Done. Reference to table 2 and its respective explanation is added in section 2.
Point 13: The code for the selection of skeletons within the SW is confusing. The idea, as far as I understand it, is to take a given number of evenly spaced poses from each SW. In the code, you have confusing notation (S_i vs. S-1 vs. S_i(1,i)).
Response 13: Thank you for this remark. The reviewer is right regarding how to build each SW. We have modified the code to clarify this issue.

Reviewer 2 Report
The paper presents a vision-based algorithm for activity recognition. The main contribution of the study is to use human skeleton pose estimation as a feature extraction method for activity detection in videos. The use of this method allows the detection of activities of multiple people in the same scene. The algorithm is also capable of classifying multi-frame activities for those that need more than one frame to be detected. The method is evaluated with the public UP-FALL dataset and compared to similar algorithms using the same dataset.
This paper presents work in an important field and to the best of my knowledge, the paper is original and unpublished.
In the related work section, the paper should be devoted to give a comprehensive review of literature, papers on rehabilitation may also be included. Recent articles are available:
Evaluation of a Rehabilitation System for the Elderly in a Day Care Center, Information 2019, 10, 3.
Regarding the structure of the paper, I would suggest to the authors to follow a more traditional structure for their paper. For example:
- Introduction
- Related work
- Materials and Methods
- Results or Experimental Results
- Discussion and Conclusion or Conclusion
In the section 6 you can introduce in 4-5 lines the chapter before starting the 6.1 section as you made in the section 5.
A discussion section Is necessary to discuss the results in detail.
Author Response
Response to Reviewer 2 Comments
Comments:
The paper presents a vision-based algorithm for activity recognition. The main contribution of the study is to use human skeleton pose estimation as a feature extraction method for activity detection in videos. The use of this method allows the detection of activities of multiple people in the same scene. The algorithm is also capable of classifying multi-frame activities for those that need more than one frame to be detected. The method is evaluated with the public UP-FALL dataset and compared to similar algorithms using the same dataset.
This paper presents work in an important field and to the best of my knowledge, the paper is original and unpublished.
Response 1: Thank you very much for these comments.
Point 1: In the related work section, the paper should be devoted to giving a comprehensive review of the literature, papers on rehabilitation may also be included. Recent articles are available:
Evaluation of a Rehabilitation System for the Elderly in a Day Care Center, Information 2019, 10, 3.
Response 1: Thank you for the suggestion. The paper has been included in the references.
Point 2: Regarding the structure of the paper, I would suggest to the authors follow a more traditional structure for their paper. For example:
- Introduction
- Related work
- Materials and Methods
- Results or Experimental Results
- Discussion and Conclusion or Conclusion
Response 2: Regarding this suggestion, we believe that the article has a structure similar to that proposed by the reviewer. Sections 4 and 5 correspond to the methodology, but it was decided to split them into 3 sections to ease the understanding of the work.
Point 3: In section 6 you can introduce in 4-5 lines the chapter before starting the 6.1 section as you made in section 5.
Response 3: Thank you for this suggestion. We have added a comment about this in section 6.
Point 4: A discussion section Is necessary to discuss the results in detail.
Response 4: We believe that a discussion section is not necessary because the results are discussed in detail in section 6 of each experiment carried out, which is reflected in the tables and figures with their respective explanations.

Reviewer 3 Report
The increased requirements to the AAL systems for monitoring and prevention of dangerous situations in the elderly imply the search for new solutions aimed at improving their reliability. Among the many parameters, events and situations that are considered important for detection and / or monitoring, the highest priority levels are given to the detection of falls and the fear of falling. This paper presents a method for fall detection and activity recognition using conventional RGB camera and estimating the pose of a person in sequence of frames. The proposed approach and the achieved results are based on previous experience of the authors in fall detection and activity recognition analyzing each frames independently. The main contribution of this work is the use of a skeleton that correctly traces the joints in-time, and then determines a person's pose by processing of the features of these skeletons in frames sequences. The presented method has been tested with different datasets and machine learning algorithms and the achieved results exceed those published in other similar studies.
The abstract is clear and correctly presents the content of the article. The reference sources correspond to the content and are cited on the appropriate places in the text.
Remarks, questions and comments:
- A part of the text ln54 - ln62 is not in English;
- The comment on the contributions of this work (ln92 - ln11) should be at the end of the article in the "Discussion" or "Conclusion" sections.
- It is not clear where the cited reference source [13] is published.
- Despite the extremely high accuracy, precision, recall, specificity, it should be noted that they are achieved with databases including records of relatively "standardized" activities and falls. For a real assessment of feasibility, it would be appropriate to perform several tests in a real environment and by simulating frequently occurring incidents, e.g. falling from a bed or chair (relatively static skeleton); falling behind a barrier in which part of the body becomes invisible when falling (lack of data on parts of the skeleton in sequence of frames); atypical gait and stumbling (unstable pose), etc.
Author Response
Response to Reviewer 3 Comments
Comments:
The increased requirements for the AAL systems for monitoring and prevention of dangerous situations in the elderly imply the search for new solutions aimed at improving their reliability. Among the many parameters, events and situations that are considered important for detection and/or monitoring, the highest priority levels are given to the detection of falls and the fear of falling. This paper presents a method for fall detection and activity recognition using a conventional RGB camera and estimating the pose of a person in a sequence of frames. The proposed approach and the achieved results are based on the previous experience of the authors in fall detection and activity recognition by analyzing each frame independently. The main contribution of this work is the use of a skeleton that correctly traces the joints in time, and then determines a person's pose by processing the features of these skeletons in frame sequences. The presented method has been tested with different datasets and machine learning algorithms and the achieved results exceed those published in other similar studies.
The abstract is clear and correctly presents the content of the article. The reference sources correspond to the content and are cited in the appropriate places in the text.
Response: Thank you very much for these comments.
Point 1: A part of the text ln54 - ln62 is not in English;
Response 1: Thank you for this comment. It was a mistake with the Latex source of the document. It has been fixed.
Point 2: The comment on the contributions of this work (ln92 - ln11) should be at the end of the article in the "Discussion" or "Conclusion" sections.
Response 2: We appreciate the suggestion, but we think that the contributions should be indicated at the beginning of the article to clarify to the reader the impact of the work.
Point 3: It is not clear where the cited reference source [13] is published.
Response 3: Thank you very much for the clarification. It was a formatting error in the reference that has already been corrected (it is a conference that was incorrectly referenced as a journal paper).
Point 4: Despite the extremely high accuracy, precision, recall, and specificity, it should be noted that they are achieved with databases including records of relatively "standardized" activities and falls. For a real assessment of feasibility, it would be appropriate to perform several tests in a real environment and by simulating frequently occurring incidents, e.g. falling from a bed or chair (relatively static skeleton); falling behind a barrier in which part of the body becomes invisible when falling (lack of data on parts of the skeleton in a sequence of frames); atypical gait and stumbling (unstable pose), etc.
Response 4: Thank you for this remark. We do agree with the reviewer, but according to the state of the art review, it was not possible to find public and tagged databases with videos in real environments for fall detection and activity recognition. Therefore, our method has been applied to public databases that have been used by other researchers to evaluate different activity recognition methods. Undoubtedly, for an evaluation closer to reality with the characteristics indicated by the reviewer, there is a pending aspect in this field of research as we mentioned in future works.

Round 2
Reviewer 1 Report
Review – round 2:
Weaknesses:
Point 1: The proposed method based on sliding windows is quite trivial: instead of a single pose, the authors add additional poses to the ML input. The only slightly advanced idea is to select a limited number of poses from each sliding window. However, I did not understand how the parameters W and S were selected.
Response 1: Thank you for this comment. The window size (W) and the number of skeletons (S) are selected from the best candidate found in the exhaustive search, which is located at W=2 and S=3. This is explained in subsection 4.4.
Reviewer comment: ok
Point 2: What is the recall that is maximized by an exhaustive search?
Response 2: Thank you for this comment. The recognition recall of the 12 UP-Fall activities is maximized by using sliding windows with skeleton sequences instead of performing the frame-by-frame analysis proposed in [12]. This is explained in subsection 4.4.
Reviewer comment: ok
Point 3: It is quite surprising that the optimal result was obtained for just 2 skeletons per SW. Some analysis of this behaviour would be good.
Response 3: Please note that the best result is obtained with 3 skeletons. Please also note that since the exhaustive search is performed with the UP-Fall database, using 3 skeletons works well for this particular case, and it may be different for other databases.
Reviewer comment: Sorry, my mistake, the best result is indeed for 3 skeletons. However, as you correctly point our, this may be specific for a given dataset. I suggest cross-validating the choice of W and S over the UR-fall dataset.
Point 4: Was this setting used in all the experiments in Section 6?
Response 4: Yes, we have used the same setting in all experiments. We have added a comment in section 6.
Reviewer comment: ok
Point 5: No real comparison to state-of-the-art is made except for a comparison with authors’ previous work that worked with isolated poses. It is no surprise that the sequences work better – there is more information available. The authors compare their results to several other works in Section 6.3, but a comparison of accuracy over different datasets makes no sense.
Response 5: Thank you very much for the remark. The idea is to show other works that have managed to recognize actions with skeleton sequences. We have modified the text to clarify this point in Table 6. On the other hand, using 3 skeletons and choosing the first frame, the middle frame and the last frame makes sense to recognize an action that occurs in the window. The fact that the system works better with less data is also related to the concept of the Curse of dimensionality (see [*, **] below). This is clarified in subsection 4.4.
* Verleysen, M., & François, D. (2005, June). The curse of dimensionality in data mining and time series prediction. In International work-conference on artificial neural networks (pp. 758-770). Springer, Berlin, Heidelberg
** Bengio, Y., Delalleau, O., & Le Roux, N. (2005). The curse of dimensionality for local kernel machines. Techn. Rep, 1258, 12.ISO 690
Reviewer comment: I believe that you should better explain that you are dealing with two tasks in your work – fall detection and action recognition (I explain this issue further in response to Point 10). So your point here is that while there are other works performing action recognition, yours works better. You believe that this is caused by the relative simplicity of the model, which is not prone to overfitting and curse of dimensionality. Do I get it right? Please, explain this better in the paper.
Point 6: On the other hand, relevant works that use sequences of poses over the same dataset (e.g, [9], [16]) are not mentioned in the comparison.
Response 6: Thank you for this remark. We have modified the text to add this analysis in section 6.2 and Table 4. It is possible to see that our approach outperforms other works. The comparison has been done with [16] (now [17)) and [32]. The reference [9] (now [10]) uses UR-FALL instead of UP-FALL such as our work and [17,32,12].
Reviewer comment: Table 4 is fine, thanks. However, you should do a similar comparison also for the UR-FALL dataset (i.e., compare best existing results on this dataset to yours).
Point 7: I find it extremely suspicious that an LSTM model evaluated by the authors worked poorly. Other works have found LSTM models very good on similar tasks. Without a strong explanation of this behaviour, I am more inclined to think that the reported LSTM implementation has some problems. I believe that LSTMs (or potentially other recursive NN models) are the strongest competitors of the proposed approach and should be thoroughly discussed in the paper.
Response 7: We do agree with the reviewer, it was also a surprise for us that the LSTM didn't work well. Several tests were carried out for the recognition of multiple activities. Different LSTM configurations with various densities and hyperparameters were tried. We hypothesise that there is too much-unbalanced information, which normally causes overfitting in the models. For example see classes 1, 2, 3, 4, 5, 9 and 12, which contain less than 1% of the total 203.525 SWs as mentioned in the second paragraph of section 6.3.
Reviewer comment: Ok, I am probably beginning to understand. LSTM works fine for fall detection (as reported in [17]) but not for action recognition, right? At least for the particular UP-Fall dataset, where there are unbalanced classes. Please, explain this more clearly.
Point 8: The costs of ML training with the sequences of skeletons are never considered.
Response 8: Thank you very much for this remark. The costs of training ML models with the skeleton sequences were added at the end of section 6 (see Table 8).
Reviewer comment: Table 8 is fine but I suggest to expand it with information about training times for single-pose features (i.e., the baseline approach from [12]). As regards the prediction time, is there any noticeable difference between individual methods and between the baseline and sw-based methods?
Point 9: The text needs to be thoroughly revised. Apart from a paragraph in Spanish (middle of page 2), there are many repetitions of the same information (e.g., the list of contributions from page 3 is very much repeated on page 7).
Response 9: Thank you very much for this suggestion. We have revised the whole article.
Reviewer comment: Unfortunately, you haven’t revised the whole article. There are still some Spanish sentences on page 2, the repetitions are still there (in multiple places; please read again the abstract and Introduction, the information is repeated, the text does not flow nicely).
Point 10: It is vital to clearly explain the task that is being studied – is it action detection (finding a specific action, e.g. fall, within a long motion sequence/stream) or action recognition (deciding the type of a short motion piece that was manually segmented)? How is the input motion processed? Sliding windows are used, so multiple windows will be evaluated per one input. How are the results combined?
Response 10: The focus of the work is activity recognition as indicated in the title. The purpose is to recognize different actions in a video that lasts up to 4 minutes. All frames of the video are labeled. One sliding window (SW), which contains 3 frames, is associated with the most frequent (statistical mode) of the three labels. This is explained in section 3.2.
Reviewer comment: I still believe that the tasks you are solving are not sufficiently specified in the paper. It should be clear from the start that there are two different tasks – fall detection and action recognition. Each of them is evaluated per-frame, right? So there is no voting of frames after the detection, the software does not produce long qualifying sequences. It produces per-frame decisions about classes. The precision, recall etc. are evaluated on the level of individual frames.
Consider restructuring section 2, adding the description of tasks to the beginning. Also, I suggest moving the description of UP-fall and UR-fall (currently in sections 6.2 and 6.4) to section 2, as well as the definition of quality metrics (section 5). Are there any interesting differences between UP-fall and UR-fall that can cause the ML methods to behave differently?
Also, you are often referencing your previous work [12] for details about the whole fall detection/action recognition pipeline. The current paper should be self-explanatory. Please restructure the first part of section 3 so that you first explain the pipeline of [12] and then introduce the refinements of the current paper. The beginning of section 5 should be moved here.
Point 11: Give more attention to explaining the basic principles of the proposed methods, instead of delving fast into detailed descriptions.
Response 11: Thank you for this remark. We think that the principles of the approach are explained in detail in Figure 1 (section 3). However, we have added a new sentence to clarify this point in section 3.
Reviewer comment: This comment was related to the previous one – I feel that the readability of the whole text can be much improved by better structuring.
Point 12: Table 2 is not referenced from the text. What does it show? First, a few columns deal with dataset properties, the last two columns mention ML methods and performance. What methods are these? not the best existing approaches for given datasets (e.g., the accuracy stated for UP-Fall is very low).
Response 12: Done. Reference to table 2 and its respective explanation is added in section 2.
Reviewer comment: This is better, thanks. Still, table 2 mentions multiple methods for some datasets and 1 quality evaluation. This corresponds to what, the best of the listed methods? You should explain that and highlight the respective method. Also, the column Trials is rather difficult to interpret. I would like to see some comparable properties of the datasets, such as the total number of classes, total number of actions, total size of dataset. Some of this can be derived from columns Fall types and Trials but not for all datasets.
Point 13: The code for the selection of skeletons within the SW is confusing. The idea, as far as I understand it, is to take a given number of evenly spaced poses from each SW. In the code, you have confusing notation (S_i vs. S-1 vs. S_i(1,i)).
Response 13: Thank you for this remark. The reviewer is right regarding how to build each SW. We have modified the code to clarify this issue.
Reviewer comment: You have added the full python code. This is not easy to read. Please try to write a simple and clear pseudocode. Something like this:
featureMatrix = new Matrix[?,?]
for window_size in {0.5,1,2} do
for s in {1,FPS} do
for i in {1,len(video) – window_size*FPS} do
full_SW = …
reduced_SW = s evenly spaced frames from full_SW
//add reduced_SW to the correct place in featureMatrix
done
done
done
Author Response
Response to Reviewer 1 Comments
Point 3: It is quite surprising that the optimal result was obtained for just 2 skeletons per SW. Some analysis of this behaviour would be good.
Reviewer comment 3: Sorry, my mistake, the best result is indeed for 3 skeletons. However, as you correctly point out, this may be specific for a given dataset. I suggest cross-validating the choice of W and S over the UR-fall dataset.
Response: Thank you for this comment. This idea that you propose, we have already developed before as the values of W and S (by exhaustive search) were obtained from the optimization using UP-Fall, which were later validated with the UR-Fall Database as shown in Table 7. It would also be possible to do it the other way around, i.e. find optimal W and S values for UR-Fall and validate them with UP-Fall, however, this would take about a week.
Point 5: No accurate comparison to state-of-the-art is made except for comparison with authors’ previous work that worked with isolated poses. It is no surprise that the sequences work better – there is more information available. The authors compare their results to several other works in Section 6.3, but a comparison of accuracy over different datasets makes no sense.
Reviewer comment 5: I believe that you should better explain that you are dealing with two tasks in your work – fall detection and action recognition (I explain this issue further in response to Point 10). So your point here is that while there are other works performing action recognition, yours works better. You believe that this is caused by the relative simplicity of the model, which is not prone to overfitting and the curse of dimensionality. Do I get it right? Please, explain this better in the paper.
Response: Regarding the difference between detection and recognition activities, we use the same approach (classification frame by frame of a video) to perform two different tasks: bi-classification (fall/not fall) and multi-classification (classification of 12 activities). We have modified the introduction to clarify this point.
Point 6: On the other hand, relevant works that use sequences of poses over the same dataset (e.g, [9], [16]) are not mentioned in the comparison.
Reviewer comment 6: Table 4 is fine, thanks. However, you should do a similar comparison also for the UR-FALL dataset (i.e., compare the best existing results on this dataset to yours).
Response: Thank you for this comment, we have added Table 8 to clarify this issue.
Point 7: I find it extremely suspicious that an LSTM model evaluated by the authors worked poorly. Other works have found LSTM models very good on similar tasks. Without a strong explanation of this behaviour, I am more inclined to think that the reported LSTM implementation has some problems. I believe that LSTMs (or potentially other recursive NN models) are the strongest competitors of the proposed approach and should be thoroughly discussed in the paper.
Reviewer comment 7: Ok, I am probably beginning to understand. LSTM works fine for fall detection (as reported in [17]) but not for action recognition, right? At least for the particular UP-Fall dataset, where there are unbalanced classes. Please, explain this more clearly.
Response: Thank you for this remark. We believe that the main problem to get better results with LSTM is related to unbalanced classes. The classification problem for fall/not fall could work fine with LSTM because the data in our case for the two classes is much more balanced (78% for not falling versus 21% for falls). On the other hand, in the multi-classification problem, we have 7 classes with less than 1% of the data.
Point 8: The costs of ML training with the sequences of skeletons are never considered.
Reviewer comment 8: Table 8 is fine but I suggest expanding it with information about training times for single-pose features (i.e., the baseline approach from [12]). As regards the prediction time, is there any noticeable difference between individual methods and between the baseline and SW-based methods?
Response: Thank you very much for this comment. The sequences method requires the classification of three frames to output a class, which means that the prediction time is greater than the baseline approach which uses only one frame.
Point 9: The text needs to be thoroughly revised. Apart from a paragraph in Spanish (middle of page 2), there are many repetitions of the same information (e.g., the list of contributions from page 3 is very much repeated on page 7).
Reviewer comment 9: Unfortunately, you haven’t revised the whole article. There are still some Spanish sentences on page 2, and the repetitions are still there (in multiple places; please read again the abstract and Introduction, the information is repeated, and the text does not flow nicely).
Response: Thank you for these remarks and comments. Unfortunately, some uncomment Spanish text was not deleted. We have done our best to improve the readability of the paper.
Point 10: It is vital to clearly explain the task that is being studied – is it action detection (finding a specific action, e.g. fall, within a long motion sequence/stream) or action recognition (deciding the type of a short motion piece that was manually segmented)? How is the input motion processed? Sliding windows are used, so multiple windows will be evaluated per one input. How are the results combined?
Reviewer comment 10: I still believe that the tasks you are solving are not sufficiently specified in the paper. It should be clear from the start that there are two different tasks – fall detection and action recognition. Each of them is evaluated per frame, right? So there is no voting of frames after the detection, and the software does not produce long qualifying sequences. It produces per-frame decisions about classes. The precision, recall etc. are evaluated on the level of individual frames.
Consider restructuring section 2, adding the description of tasks to the beginning. Also, I suggest moving the description of UP-fall and UR-fall (currently in sections 6.2 and 6.4) to section 2, as well as the definition of quality metrics (section 5). Are there any interesting differences between UP-fall and UR-fall that can cause the ML methods to behave differently?
Also, you are often referencing your previous work [12] for details about the whole fall detection/action recognition pipeline. The current paper should be self-explanatory. Please restructure the first part of section 3 so that you first explain the pipeline of [12] and then introduce the refinements of the current paper. The beginning of section 5 should be moved here.
Response: Thank you very much for this comment.
We have modified the introduction to clarify the tasks of our work. See also our response to your point 5 for more detail about the tasks we performed. We have also modified section 3 to clarify the classification of the sliding windows. The metrics in section 5 (recall, precision, etc) are calculated considering the sliding windows.
Regarding the suggestion of restructuring section 2, we prefer to keep the present structure of the section and leave the details for section 6.
As far as we know, there is no significant difference between UP-fall and UR-fall.
Finally, we believe Figures 1 and 2 are good enough to explain the entire pipeline of our approach.
Point 11: Give more attention to explaining the basic principles of the proposed methods, instead of delving fast into detailed descriptions.
Reviewer comment 11: This comment was related to the previous one – I feel that the readability of the whole text can be much improved by better structuring.
Response: Thank you for these remarks and comments. We have done our best to improve the readability of the paper.
Point 12: Table 2 is not referenced from the text. What does it show? First, a few columns deal with dataset properties, and the last two columns mention ML methods and performance. What methods are these? not the best existing approaches for given datasets (e.g., the accuracy stated for UP-Fall is very low).
Reviewer comment 12: This is better, thanks. Still, table 2 mentions multiple methods for some datasets and 1 quality evaluation. This corresponds to what, the best of the listed methods? You should explain that and highlight the respective method. Also, the column Trials is rather difficult to interpret. I would like to see some comparable properties of the datasets, such as the total number of classes, total number of actions, and total size of the dataset. Some of this can be derived from columns Fall types and Trials but not for all datasets.
Response: Thank you very much for your comment. As we have mentioned before, the evaluation corresponds to the best result of the original work. We highlighted the best method. The trials column (the fourth column) is described in the text. Regarding the suggestion of including more columns, we appreciate the comment, but unfortunately, we are not able to do it given the short time to respond to this letter.
Point 13: The code for the selection of skeletons within the SW is confusing. The idea, as far as I understand it, is to take a given number of evenly spaced poses from each SW. In the code, you have confusing notation (S_i vs. S-1 vs. S_i(1,i)).
Reviewer comment 13: You have added the full python code. This is not easy to read. Please try to write simple and clear pseudocode. Something like this:
featureMatrix = new Matrix[?,?]
for window_size in {0.5,1,2} do
for s in {1,FPS} do
for i in {1,len(video) – window_size*FPS} do
full_SW = …
reduced_SW = s evenly spaced frames from full_SW
//add reduced_SW to the correct place in featureMatrix
done
done
done
Response: Thank you very much for this comment. Done.
